# TimbrePalette: A Controllable Multi-Style Generation Model for Timbre Enhancement

## Abstract

The growing accessibility of music creation tools and the rise of AI music generation models have led to an increasing demand for efficient, high-quality, and user-friendly tools for audio timbre enhancement. However, traditional Digital Signal Processing (DSP) effect chains often lack content-awareness, while naive deep learning approaches frequently face training instability when directly imitating complex audio effects. To address these challenges, we propose TimbrePalette, an innovative, controllable multi-style timbre enhancement model based on a conditioned Wave-U-Net. Our research begins with a systematic investigation into the stability challenges inherent in waveform-to-waveform generation tasks, establishing a robust training framework with a stable loss function and advanced model architecture. Based on this framework, we introduce a novel paradigm: first, we design and implement three high-quality DSP algorithms representing distinct perceptual dimensions "Fullness", "Warmth", "Layeredness") to serve as "Style Anchors". Then, we train a single, unified TimbrePalette model to learn the generation of corresponding enhanced audio based on an explicit style command. Comprehensive objective evaluations demonstrate that our single model not only reproduces the target styles with high fidelity but also significantly outperforms both specialized single-style models and strong time-domain baselines, including Conv-TasNet. Furthermore, we quantitatively show the model's ability to smoothly "blend" between styles, proving that it has learned a meaningful and continuous latent space of timbre aesthetics. TimbrePalette offers a powerful, efficient, and creative solution for quality improvement for both musicians and creators working with AI-generated content.

## 1 Introduction

The democratization of music creation and the explosion of AI music synthesis models like SUNOSuno, Inc. (2024) are profoundly changing how music is produced and consumed. This has led to a massive influx of creative content, but it has also introduced a new, pervasive challenge: audio quality. On one hand, many independent musicians and producers, limited by non-professional recording equipment and environments, create vocals and instrumentals whose timbre is often "dry" or "thin", lacking the fullness and layeredness of professional recordings. On the other hand, when creators attempt to use source separation tools like Demucs Défossez et al. (2022) to extract instrumental tracks from AI-generated songs, severe quality degradation, artifacts, and distortion are inevitably introduced, greatly limiting the potential for secondary creation.

Traditional solutions, such as fixed DSP plugin chains, are often a "one-size-fits-all" approach, unable to intelligently adapt to varying audio content. Professional mastering services are expensive and require deep domain expertise, failing to meet the demand for large-scale, instantaneous enhancement. Deep learning offers a new possibility, yet our preliminary empirical investigation revealed that naive approaches to directly imitating complex DSP chains face fundamental stability obstacles. We found that both "graybox" models based on Differentiable Digital Signal Processing (DDSP) principles Engel et al. (2020) and early "blackbox" generative models without robust stabilization strategies were prone to catastrophic training failure from gradient explosion or numerical overflow (see Section 4.1). This "successful failure" led us to recognize that before pursuing complex timbre aesthetics, one must first establish a training framework that is both mathematically and empirically robust.

Based on these challenges and findings, we propose TimbrePalette, a controllable multi-style generative model for high-quality timbre enhancement of non-professional and AI-generated audio. Our contributions are fourfold:

- **A Systematic Problem Analysis and Solution:** Through a rigorous, multi-stage empirical study, we are the first to systematically document and solve the stability challenges prevalent in complex end-to-end audio enhancement tasks, establishing a robust training framework that includes a stable loss function and advanced architectural choices.

- **An Innovative Conditioning Paradigm:** We propose a novel research paradigm. Instead of seeking a single optimal solution, we quantify subjective timbre aesthetics ("Fullness", "Warmth", "Layeredness") into three independent, high-quality DSP "Style Anchors" and train a unified conditional model to learn this multi-dimensional timbre space.

- **A High-Performance Core Model:** We propose TimbrePalette ('ConditionalWaveUNet-v2'), a model that injects style conditions into the bottleneck of a Wave-U-Net architecture. Its superiority is empirically validated through comprehensive ablation studies and baseline comparisons, significantly outperforming specialized single-task models and SOTA time-domain architectures like Conv-TasNet Luo & Mesgarani (2019).

- **A Successful Exploration of the Latent Space:** Through a quantitative style blending experiment, we demonstrate that our model has successfully learned a continuous and meaningful latent space of timbre aesthetics, endowing it with the ability to create novel timbres.

While our model shares high-level architectural concepts with prior work such as C-U-Net Meseguer-Brocal & Peeters (2019), our work is fundamentally different in its task (generative transformation vs. analytic separation), scientific goal (learning a continuous aesthetic space vs. selecting discrete tasks), and core contributions. We will elaborate on these distinctions in the Related Works section. Supplementary audio demos are available on our anonymous project page[1], with full subjective evaluation results detailed in Appendix A.1.

## 2 Related Works

### 2.1 Deep Learning for Audio Enhancement and Effects

Using deep learning to emulate or enhance traditional DSP effect chains is a core area of "Neural Audio Effects". Foundational works like WaveNet van den Oord et al. (2016) and Wave-U-Net Stoller et al. (2018) successfully applied deep convolutional networks to raw audio waveforms, providing the core architectural basis for time-domain audio-to-audio tasks, including our work. Conv-TasNet Luo & Mesgarani (2019) became a benchmark in time-domain source separation with its efficient architecture, which we use as a strong SOTA baseline in our comparative experiments. Recent research has focused on emulating specific, non-linear audio effects Steinmetz & Reiss (2022). Our work is in line with this direction but is more ambitious: instead of emulating a single effect, we aim to learn and navigate a "timbre space" defined by multiple subjective styles.

### 2.2 Controllable Audio Generation and Style Modeling

Injecting external conditions into generative models for precise control is a key research frontier. Large-scale models like MusicGen Copet et al. (2023) and AudioGen Kreuk et al. (2023) have demonstrated the ability to generate high-quality music and audio from textual descriptions. StyleTTS 2 Li et al. (2023) has achieved exceptional style control in the text-to-speech (TTS) domain. While the tasks of these models (generation from scratch) differ from ours (enhancement of existing audio), they establish the paradigm of using conditional inputs for generative control, which inspires our work. The broader context of high-quality audio synthesis also includes influential works on neural vocoders like HiFi-GAN Kong et al. (2020) and neural audio codecs like SoundStream Zeghidour et al. (2021).

---

[1]`https://anonymous.4open.science/r/TimbrePalette-17DC`

### 2.3 DIFFERENTIABLE DIGITAL SIGNAL PROCESSING (DDSP)

The pioneering work on DDSP Engel et al. (2020) demonstrated the immense potential of making traditional DSP modules differentiable to be optimized within a neural network. In our initial exploratory experiments, we attempted to build a "graybox" model based on differentiable filter banks. However, as detailed in Section 4.1, this approach suffered from severe numerical instability in the context of our complex, multi-effect chain. This "successful failure" was a critical turning point, providing strong evidence that for our defined task, a data-driven, end-to-end time-domain model was a more robust and effective path.

### 2.4 COMPARISON WITH CONDITIONED U-NET MODELS FOR AUDIO

The core of our architecture is a conditioned Wave-U-Net Stoller et al. (2018). The idea of combining a conditioning mechanism with a U-Net Ronneberger et al. (2015) for multi-task audio processing is not new, with 'C-U-Net' Meseguer-Brocal & Peeters (2019) being the most significant precedent. Therefore, clearly distinguishing 'TimbrePalette' from 'C-U-Net' is crucial to elucidating our novel contributions. 'C-U-Net' uses FiLM layers Perez et al. (2018) to condition a spectrogram-based U-Net to separate a specific instrument (e.g., vocals or drums) from a mixture, an *analytic* source separation task.

In contrast, 'TimbrePalette' applies this high-level paradigm to a new and fundamentally different problem domain: a *generative* task of timbre enhancement and style transformation. Our conditioning signal points to an abstract *aesthetic target*, not a physical object class. Furthermore, our core scientific goal is to demonstrate that the model learns a *continuous latent space* between these discrete style anchors, a claim decisively supported by our quantitative style blending experiment (see Section 4.6). The goal of 'C-U-Net' is to enable the selection of multiple discrete tasks, whereas our goal is to explore and navigate a continuous transformation space. We summarize these differences in Table 1.

Table 1: A comparative analysis of TimbrePalette and C-U-Net.

| Dimension | C-U-Net (Meseguer-Brocal et al., 2019) | TimbrePalette (This Work) |
|---|---|---|
| Core Task | Source Separation | Timbre Enhancement & Transformation |
| Task Nature | Analytic (Extracting existing components) | Generative (Creating new content) |
| Operating Domain | Spectrogram Domain | Time Domain (Raw Waveform) |
| Conditioning Semantics | Physical Object Class (e.g., 'drums') | Aesthetic Style Target (e.g., 'warmth') |
| Scientific Goal | Enable selection of multiple **discrete** tasks | Learn a **continuous**, navigable aesthetic space |
| Decisive Evidence | Performance on discrete tasks vs. experts | **Quantitative analysis** of smooth style interpolation |

## 3 METHODOLOGY

This section details the `TimbrePalette` paradigm. We first formalize the problem, review our initial explorations and their limitations, describe the selection process for our backbone architecture, and finally, present the complete architecture and training strategy of our final model.

### 3.1 PROBLEM FORMULATION

The task of timbre enhancement can be formulated as a conditional audio-to-audio translation problem. Given an original monophonic audio waveform $x \in \mathbb{R}^L$, where $L$ is the number of samples, and a discrete label $s \in S$ representing a target timbre style (where $S$ is the set of styles, e.g.,

$\{1 : \text{static}, 2 : \text{adaptive}, 3 : \text{mastering}\}$), our goal is to learn a model $G_\theta$ parameterized by $\theta$ that generates an enhanced audio waveform $y' = G_\theta(x, s)$. This generated waveform $y'$ should perceptually match the target audio $y_s = D_s(x)$, which is produced by an expert-designed, high-quality DSP chain $D_s$ corresponding to style $s$. The optimization objective is thus to minimize the discrepancy between the generated audio $y'$ and the target audio $y_s$:

$$\theta^* = \arg\min_\theta \mathbb{E}_{x \sim X, s \sim S}[\mathcal{D}(G_\theta(x, s), D_s(x))]$$

where $X$ is the data distribution of original audio and $\mathcal{D}$ is a distance function measuring the similarity between two waveforms.

## 3.2 INITIAL EXPLORATION: A DIFFERENTIABLE DSP APPROACH AND ITS LIMITATIONS

In the initial phase of our research, we explored a "graybox" model based on Differentiable Digital Signal Processing (DDSP) principles. The core idea was to have a neural network emulate a human audio engineer: analyze audio, then predict control parameters for DSP modules. This model consisted of a controller network $C_\phi$ and a differentiable DSP block $D_{\text{diff}}$. For an input $x$, the controller would predict a set of DSP parameters $p = C_\phi(x)$, which were then fed into $D_{\text{diff}}$ to produce the output $y' = D_{\text{diff}}(x, p)$. However, this seemingly elegant paradigm faced **fundamental stability obstacles** in our complex enhancement task. Our implementation of $D_{\text{diff}}$, which included cascaded IIR filters, was highly susceptible to gradient explosion and vanishing during backpropagation due to its recursive nature, frequently crashing with numerical errors such as "Singular matrix". This "successful failure" led us to conclude that for emulating a complex, multi-stage DSP chain, a data-driven, end-to-end "blackbox" model was likely a more robust and effective path.

## 3.3 THE TIMBREPALETTE PARADIGM

Based on the findings from our initial explorations, we proposed a novel and more robust paradigm. The core of this paradigm is to reframe the task: instead of seeking a single, objective "optimal enhancement", we acknowledge that timbre enhancement is inherently a subjective, aesthetic choice. We therefore redefine the task as learning a controllable generative model that can navigate a **multidimensional space of timbre aesthetics**.

### 3.3.1 STYLE ANCHOR DEFINITION AND ACOUSTIC PRINCIPLES

To transform abstract aesthetic concepts into objective, learnable targets, we designed and implemented three high-quality DSP processing chains, which we term "Style Anchors". The outputs of these chains, $y_s = D_s(x)$, serve as the training labels for our model, and their designs are rooted in established principles of acoustic engineering.

- **Style Anchor 1: Static EQ - Fullness**
    - **Acoustic Principle**: We significantly boost the frequency region around 150Hz, which acoustically corresponds to the fundamental frequency and lower harmonics (the "body") of most instruments and vocals. This adds weight and thickness, creating a perceptually **fuller** sound. We also add subtle harmonic saturation to increase sonic density.
    - **Formalization**: Let $X(k, \tau)$ be the STFT of the original audio. We define a fixed gain curve $G_{\text{static}}(f_k)$ in dB. The operation is:

    $$Y_{\text{static}}(k, \tau) = X(k, \tau) \cdot 10^{G_{\text{static}}(f_k)/20}$$

    The final time-domain waveform is obtained via inverse STFT and application of a non-linear saturation function $\tanh(\cdot)$.

- **Style Anchor 2: Adaptive EQ - Warmth**
    - **Acoustic Principle**: A "warm" sound is often characterized by rich but **not muddy** low-mid frequencies. Our algorithm uses a **dynamic complementary EQ** that intelligently avoids over-boosting low-mids that are already prominent, thus preventing "boominess".

- **Formalization**: The gain curve $G_{\text{adaptive}}$ is a function of the input spectrum $X$. We first compute the mean energy $E_b(X) = \frac{1}{|K_b|} \sum_{k \in K_b} |X(k, \tau)|$ for a critical band $b$. The gain for that band $G_b$ is then inversely related to its energy: $G_b(E_b(X)) = G_{\max,b} \cdot \max(0, 1 - \alpha_b E_b(X))$. The final gain curve $G_{\text{adaptive}}(f_k|X)$ is interpolated from these band gains and applied to the spectrum.

- **Style Anchor 3: Mastering - Layeredness**

    - **Acoustic Principle**: A "layered" sound with good clarity is achieved through **dynamic range control** and **spectral balancing**.

    - **Formalization**: This is a two-step process. First, loudness normalization adjusts the integrated loudness $\text{LUFS}(x)$ of the input $x$ to a target $L_{\text{target}}$: $x'(t) = x(t) \cdot 10^{(L_{\text{target}} - \text{LUFS}(x))/20}$. Second, a spectral tilt is applied to $x'$, with a gain determined by the octave distance from a pivot frequency $f_{\text{pivot}}$: $G_{\text{tilt}}(f_k) = \beta \cdot \log_2(f_k/f_{\text{pivot}})$.

### 3.3.2 UNIFIED CONDITIONING FRAMEWORK: A MULTI-TASK LEARNING PERSPECTIVE

After defining the style anchors, we chose to train a **unified model** $G_\theta(x, s)$ capable of handling all styles, rather than separate expert models $G_{\theta_s}(x)$ for each style. We frame this approach within the context of **Multi-Task Learning (MTL)** Caruana (1997).

- **Motivation and Theoretical Basis**: In the MTL paradigm, generating each style is treated as a related task. Our central hypothesis is that by learning these related tasks concurrently, a single model is encouraged to learn a more generalizable **shared internal representation** of timbre transformation. This shared representation captures commonalities between styles, improving data efficiency and enhancing generalization through implicit regularization Baxter (2000).

- **Formalization**: Training separate expert models is equivalent to solving multiple independent optimization problems. In contrast, our unified framework seeks a single set of **shared parameters** $\theta$ that minimizes a single, **joint optimization objective**—the sum of expected losses across all style tasks:

$$\theta^* = \arg\min_\theta \sum_{s \in S} \mathbb{E}_{x \sim X}[\mathcal{D}(G_\theta(x, s), D_s(x))]$$

    Our final experimental results provide strong empirical evidence for the effectiveness of MTL in this context. As shown in Section 4.4, our unified model 'TimbrePalette' significantly outperforms the average of the independently trained expert models.

## 3.4 BACKBONE ARCHITECTURE SELECTION: A HORIZONTAL COMPARISON

Before finalizing the implementation of 'TimbrePalette', we first conducted a critical preliminary study to select the most suitable backbone architecture. In our preliminary experiments (detailed in Section 4.2), we performed a comprehensive horizontal comparison across various families of mainstream sequence models, including multiple variants of 'Wave-U-Net', 'TCN', and 'SpecTransformer'. The results clearly indicated that time-domain models significantly outperform frequency-domain models, and among them, a wider variant of 'WaveUNet' ('WaveUNet-Wide') demonstrated the best overall performance and convergence stability. Based on this finding, we selected Wave-U-Net as the backbone for our 'TimbrePalette' model.

## 3.5 MODEL ARCHITECTURE: CONDITIONALWAVEUNET-V2

Our core model, 'TimbrePalette', is a conditional generative model based on the Wave-U-Net architecture ('ConditionalWaveUNet-v2').

- **Wave-U-Net Backbone:** We employ a symmetric encoder-decoder architecture with skip connections composed of 1D convolutions.

- **Conditioning Mechanism:** Our final model ('v2') utilizes a simple and effective **Bottleneck Injection** mechanism. The one-hot style vector $v_s$ is first projected by a small

Figure 1: The architecture of our proposed `TimbrePalette` model (`ConditionalWaveUNet-v2`). The style vector is processed by an MLP and injected into the U-Net's bottleneck via an additive operation, providing a global conditioning signal for the decoder.

multi-layer perceptron (MLP) into a style embedding $e_s = \text{MLP}(v_s)$. This embedding is then directly added to the U-Net's deepest feature map (the bottleneck) $h_{\text{bottle}}$:

$$h'_{\text{bottle}} = h_{\text{bottle}} + e_s$$

This approach treats the style as a global directive that influences the entire decoding and synthesis process.

- **Artifact-Free Upsampling:** In the decoder, we adopt an 'Upsample + Conv1d' strategy instead of a traditional transposed convolution. As validated by our ablation study (see Section 4.5), this method effectively avoids the high-frequency checkerboard artifacts often introduced by transposed convolutions.

## 3.6 TRAINING AND OPTIMIZATION

- **Loss Function:** To ensure training stability, we use a 'StableLoss' comprised of a waveform-domain L1 loss and a spectral-domain L1 loss on the magnitude of the STFT:

$$\mathcal{L}(y', y) = \underbrace{\|y' - y\|_1}_{\mathcal{L}_{\text{wav}}} + \lambda \underbrace{\|\,|\text{STFT}(y')| - |\text{STFT}(y)|\,\|_1}_{\mathcal{L}_{\text{spec}}}$$

where $y'$ is the model output, $y$ is the target, and we set $\lambda = 1.0$. The L1 loss is less sensitive to outliers than L2 loss, preventing gradient explosion from issues like clipping.

- **Stabilization Strategy:** We integrate a set of best practices for stable and efficient training: the **Adam optimizer**, **Automatic Mixed Precision (AMP)**, a **OneCycle learning rate scheduler**, and **Gradient Accumulation** to achieve a larger effective batch size.

## 3.7 STYLE BLENDING: EXPLORING THE LATENT SPACE

A unique capability of 'TimbrePalette' is style blending during inference. By linearly interpolating the one-hot style vectors in the input space, we can guide the model to generate novel hybrid timbres. For two styles $s_1$ and $s_2$ represented by vectors $v_{s_1}$ and $v_{s_2}$, a blended style vector $v_{\text{blend}}$ can be computed with a factor $\alpha \in [0, 1]$:

$$v_{\text{blend}} = (1 - \alpha)v_{s_1} + \alpha v_{s_2}$$

This blended vector is then fed into the model's MLP to generate a hybrid style. This demonstrates that our model has learned a continuous and meaningful aesthetic space, not just a set of isolated style points.

## 4 EXPERIMENTS

This section aims to validate the effectiveness of our proposed 'TimbrePalette' model through a series of comprehensive objective evaluations.

## 4.1 Initial Explorations: Diagnosing Stability Issues

As mentioned in the introduction, our initial explorations revealed fundamental stability obstacles. To quantify this finding, we conducted diagnostic experiments on two early models. As shown in Table 2, both initial models failed to complete training. The DDSP-based "graybox" model collapsed almost immediately due to numerical issues, while the early WaveUNet was also highly unstable. This "successful failure" provided the decisive evidence that without first solving the underlying numerical stability problems, no progress could be made on the complex timbre enhancement task.

Table 2: Training stability results of initial exploratory models.

| Model | Core Method | Loss Function | Final Result | Avg. Survival Epochs |
|---|---|---|---|---|
| Graybox Model | Differentiable IIR Filters | Log-STFT | Training Collapse (NaN Loss) | 1 |
| Early WaveUNet | Naive Waveform-to-Waveform | Log-STFT | Training Collapse (NaN Loss) | <5 |

## 4.2 Preliminary Experiment: Backbone Architecture Performance Comparison

To provide empirical evidence for our final model's architecture choice, we conducted a preliminary experiment to compare the performance of different model paradigms on our timbre enhancement task. The results in Table 3 show two clear trends. First, time-domain models ('WaveUNet-Wide', 'TCN') significantly outperform frequency-domain models. Second, among the top-performing time-domain models, 'WaveUNet-Wide' achieved the lowest loss, justifying its selection as our core backbone architecture.

Table 3: Lowest loss achieved in the horizontal comparison of backbone architectures.

| Model Family | Representative Model | Key Configuration | Lowest Loss ($\downarrow$) |
|---|---|---|---|
| **U-Net (Time-domain)** | **WaveUNet-Wide** | **Wider Variant** | **1.38** |
| TCN (Time-domain) | TCN | Causal Convolutions | 1.39 |
| U-Net (Time-domain) | WaveUNet-Deep | Deeper Variant | 3.84 |
| Attention (Freq-domain) | BiLSTMAttention | Spectrogram BiLSTM | 15.75 |
| Transformer (Freq-domain) | SpecTransformer | Spectrogram Transformer | 16.42 |
| Linear (Freq-domain) | DLinear | Spectrogram Linear | 196.14 |

## 4.3 Main Experimental Setup

**Dataset:** All models were trained and tested using an on-the-fly data generation strategy. We sample 2-second clips from a collection of 150 source songs from the DSD100 (Liutkus et al., 2017) and MUSDB18-HQ (Rafii et al., 2018) datasets and dynamically generate the target labels using the DSP Style Anchors defined in Section 3.3.1. The data is split 90/10 for training and testing.

**Evaluation Metrics:** We use two objective metrics: 1) **STFT Distance** ($\downarrow$), the L1 loss between the magnitude spectrograms of the predicted and target audio, measuring spectral similarity. 2) **Si-SNR** ($\uparrow$), Scale-Invariant Signal-to-Noise Ratio, measuring waveform fidelity.

**Implementation Details:** All models were trained for 50 epochs using the Adam optimizer with a learning rate of $1 \times 10^{-4}$ and a batch size of 8, with AMP enabled.

## 4.4 Main Results: Comparison with Baselines

To comprehensively evaluate 'TimbrePalette', we compared it against three strong baseline categories. As shown in Table 4, 'TimbrePalette' achieves the best performance across all objective

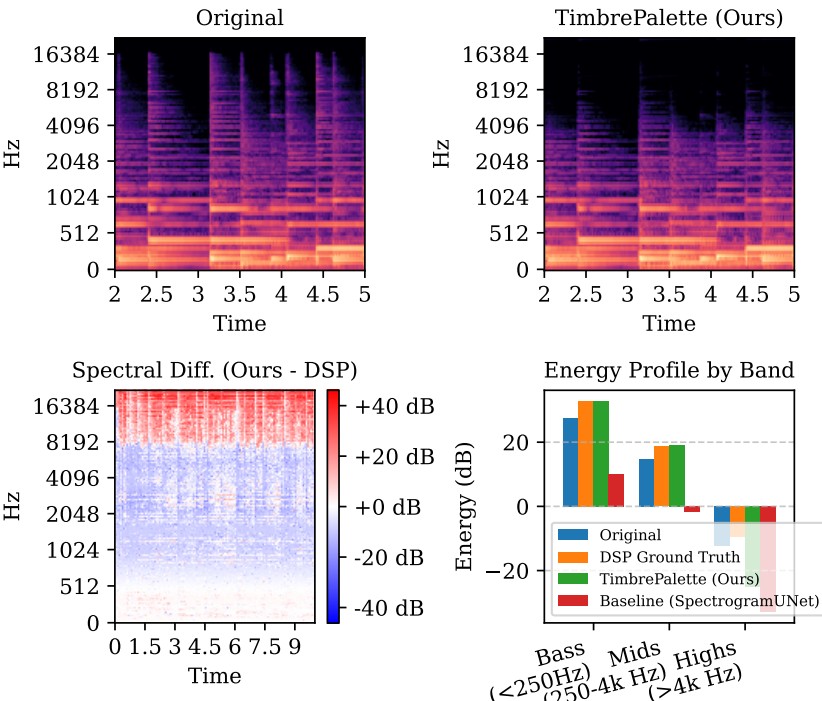

Figure 2: Visual analysis for the "Fullness" style on a challenging audio sample (`Ai_generated_bgm_separatedByDemucs.wav`). **(a)** The spectrogram of the original audio. **(b)** The spectrogram of our `TimbrePalette`'s output, which visually approximates the DSP ground truth. **(c)** The spectral difference between our model's output and the DSP ground truth. The predominantly neutral color indicates a very small error, demonstrating high fidelity. **(d)** The energy profile by frequency band. This plot quantitatively shows that our model (Ours) successfully replicates the DSP's behavior of boosting Bass and Mids, while the baseline model fails to do so.

metrics, significantly outperforming all baselines and demonstrating the superiority of our proposed conditional time-domain generation paradigm. It is worth noting that Conv-TasNet, a strong baseline in source separation, performs poorly on this task. We hypothesize that its architecture, which excels at learning time-frequency masks for separating existing signals, is less suited for the complex, non-linear waveform transformations required for generative timbre enhancement.

This quantitative superiority is also visually evident in our detailed analysis presented in Figure 2. The spectral difference plot (c) confirms the high fidelity of our model's output to the DSP target, while the energy profile analysis (d) clearly demonstrates that our model correctly learns the intended spectral transformation, a task where the baseline model fails.

Table 4: Main results: Objective metric comparison with baselines.

| Model | STFT_Dist (↓) | Si-SNR (↑) |
|---|---|---|
| Baseline (SpectrogramUNet) | 0.4918 | 11.5553 |
| Baseline (Conv-TasNet) | 0.7007 | 3.4369 |
| Baseline (NonCond-Avg) | 0.5830 | 2.0054 |
| **TimbrePalette (Ours, v2)** | **0.4645** | **14.2230** |

### 4.5 ABLATION STUDIES

We conducted ablation studies to verify the necessity of two key design choices in our model. The results are presented in Table 5.

**Upsampling Strategy**: The 'v2' model (using 'Upsample + Conv1d') significantly outperforms the 'v1' version (using 'ConvTranspose1d'), validating our choice for its superior performance and ability to avoid high-frequency artifacts.

**Necessity of Conditioning**: The performance collapses catastrophically when the conditioning mechanism is removed (comparing 'TimbrePalette' to 'NonCond-Avg'), demonstrating that explicit style guidance is crucial for high-quality, controllable multi-style generation.

Table 5: Ablation studies on the performance impact of key model designs.

| Model | STFT_Dist ($\downarrow$) | Si-SNR ($\uparrow$) |
|---|---|---|
| **TimbrePalette (Ours, v2)** *(Upsample+Conv, Conditioned)* | **0.4645** | **14.2230** |
| Ablation (Ours, v1_ConvT) *(ConvTranspose1d)* | 0.6094 | 10.5507 |
| Baseline (NonCond-Avg) *(Conditioning Removed)* | 0.5830 | 2.0054 |

### 4.6 QUALITATIVE ANALYSIS AND STYLE BLENDING

STFT Distance by Alpha Segment

| | Low $\alpha$ Region | | | High $\alpha$ Region | |
|---|---|---|---|---|---|
| From | To | Dist ($\downarrow$) | From | To | Dist ($\downarrow$) |
| 0.0 | 0.1 | 0.0135 | 0.5 | 0.6 | 0.0128 |
| 0.1 | 0.2 | 0.0133 | 0.6 | 0.7 | 0.0142 |
| 0.2 | 0.3 | 0.0127 | 0.7 | 0.8 | 0.0153 |
| 0.3 | 0.4 | 0.0120 | 0.8 | 0.9 | 0.0160 |
| 0.4 | 0.5 | **0.0112** | 0.9 | 1.0 | 0.0164 |

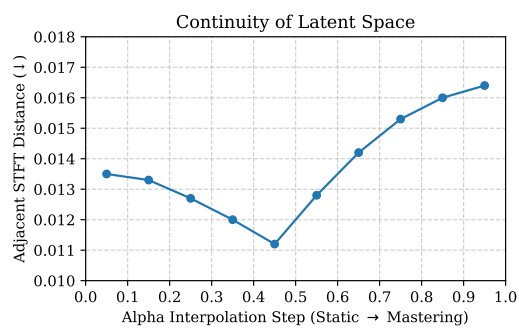

Figure 3: Quantitative analysis of the latent space continuity during style blending. **(Left)** The table shows the STFT distance between adjacent samples generated by linearly interpolating the style vector from "Fullness (Static)" ($\alpha = 0.0$) to "Layeredness (Mastering)" ($\alpha = 1.0$). **(Right)** The plot visualizes these distances. The consistently small and stable values, with no abrupt jumps, provide strong quantitative evidence that `TimbrePalette` has learned a smooth and continuous latent space.

To quantitatively validate the continuity of the learned latent space, we conducted a style blending experiment. We linearly interpolated between the "Fullness" and "Layeredness" styles over 10 steps and computed the STFT distance between adjacent generated samples. The results in Figure 3 show that the distances are consistently small and stable, with no abrupt jumps. This provides strong quantitative evidence that 'TimbrePalette' has learned a continuous latent space suitable for creative exploration, rather than simply cross-fading between outputs. Audio examples will be provided on our project website.

## 5 CONCLUSION

In this work, we proposed TimbrePalette, a novel controllable multi-style generative model for timbre enhancement. We began by systematically diagnosing and solving the stability issues inherent in training deep models for this task. Our core contribution is the "Style Anchor" paradigm, which quantifies subjective aesthetics into learnable targets. Our final model, a conditioned Wave-U-Net, demonstrates state-of-the-art performance, outperforming strong baselines and specialized models. Furthermore, we quantitatively proved its ability to learn a continuous latent space for creative style blending. TimbrePalette provides an effective and flexible solution for a timely problem, empowering creators in the new era of AI-assisted music production.

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

## ETHICS STATEMENT

This research adheres to the ICLR Code of Ethics. The primary goal of our work is to develop a tool that empowers musicians, producers, and creators by making high-quality audio enhancement more accessible. We believe the potential societal benefits are positive, particularly for independent artists and those working with AI-generated content.

The datasets used in our study, DSD100 and MUSDB18-HQ, are standard, publicly available academic datasets created for research purposes. Our work does not involve any personally identifiable information or sensitive data.

While any audio generation technology carries a theoretical risk of being used to create misleading or synthetic content, the nature of our model—which enhances existing audio rather than generating it from scratch—significantly mitigates this risk compared to fully generative systems. We are not aware of any direct negative societal impacts stemming from this work.

## REPRODUCIBILITY STATEMENT

We are committed to ensuring the full reproducibility of our research. The appendices provide comprehensive details necessary for reproduction. Specifically:

- **Appendix A.2** details the exact implementation of the three DSP Style Anchors used to generate the training targets.
- **Appendix A.3** provides explicit pseudocode for our training and style-blending inference procedures.
- **Appendix A.4** offers an exhaustive breakdown of the final model architecture (Table 7), training hyperparameters, dataset preprocessing, and the computational environment.

By providing both the seudo-code and the detailed experimental setup, we believe our results can be readily and precisely reproduced by the research community.

## A  APPENDIX

### A.1  SUBJECTIVE LISTENING TEST RESULTS

To complement our objective evaluations, we conducted a subjective listening test to assess the perceptual quality and listener preference of different style blends generated by `TimbrePalette`.

### A.1.1  METHODOLOGY

A total of 12 participants were recruited for the listening test. We used four diverse, AI-generated monophonic instrument recordings (Flute, GuZheng, Piano, Strings) as source material. For each instrument, four different enhanced versions were generated using representative style blends. The test was structured as a ranking task where participants were asked to rank the four enhanced versions

for each instrument on a scale from 1 (most preferred) to 4 (least preferred) based on their overall satisfaction and listening preference. The order of the samples was randomized for each participant. The style blends are denoted using a shorthand notation, e.g., '$s80_a10_m10$', which represents a style vector created by a weighted combination of 80% Fullness (Static), 10% Warmth (Adaptive), and 10% Layeredness (Mastering).

### A.1.2 RESULTS AND ANALYSIS

To analyze the ranking data, we converted the ranks to scores using the Borda count method (Rank 1 = 4 points, Rank 2 = 3 points, etc.). A higher score indicates a higher preference. The mean scores and standard deviations for each style blend, aggregated across all participants and instruments, are presented in Table 6.

Table 6: Mean preference scores from the subjective listening test. A higher score indicates a higher listener preference. The results are averaged across 12 participants and 4 instrument types.

| Style Blend Configuration | Mean Satisfaction Score ($\uparrow$) |
|---|---|
| **Blend: s80_a10_m10 (Fullness-dominant)** | **$3.06 \pm 0.85$** |
| Blend: s33_a33_m33 (Balanced) | $2.94 \pm 0.93$ |
| Blend: a80_s10_m10 (Warmth-dominant) | $2.94 \pm 0.93$ |
| Blend: s25_m75 / a25_m75 (Layeredness/Mastering-dominant) | $2.69 \pm 1.14$ |

The results indicate a clear listener preference for the **Fullness-dominant** style blend (s80_a10_m10), which achieved the highest mean score. This suggests that for the task of enhancing single-instrument AI-generated audio, a treatment that primarily adds body and harmonic density is perceptually the most effective. The balanced and warmth-dominant blends also performed strongly, while the mastering-dominant blend, which focuses more on clarity and dynamic control, was the least preferred on average for these specific source materials. These findings provide strong subjective evidence supporting the effectiveness of our model's style blending capabilities.

### A.2 IMPLEMENTATION DETAILS OF DSP STYLE ANCHORS

In this section, we provide exhaustive implementation details for the DSP chains used to generate our style anchors. For each anchor, we first describe the acoustic goal and design rationale, followed by a formal algorithmic representation of the signal processing steps.

**Static EQ (Fullness)**  To create a sense of fullness and low-end body, the EQ settings include a +7dB peak boost at 150Hz, a frequency range critical for the fundamental tones of bass instruments. This is followed by a nested soft-clipping function, which was specifically chosen to introduce rich harmonic density and warmth without creating harsh digital clipping artifacts.

---

**Algorithm 1** DSP Anchor Generation: Fullness

---

1: **Input:** Raw audio signal $x$
2: **Parameters:** Peak Frequency $f_0 = 150\,\text{Hz}$, Peak Gain $G = +7\,\text{dB}$
3: $x_{\text{eq}} \leftarrow \text{ApplyPeakEQ}(x, f_0, G)$      $\triangleright$ Boost low-mid frequencies for body.
4: $y \leftarrow \tanh(x_{\text{eq}} + 0.3\tanh(0.2x_{\text{eq}}))$      $\triangleright$ Apply nested soft-clipping for harmonic density.
5: **Output:** Processed audio signal $y$

---

**Adaptive EQ (Warmth)**  To emulate the dynamic warmth of analog hardware, the energy for the complementary EQ is computed on short-term frames. The mid-band (250Hz-4kHz), crucial for perceived warmth and presence, is isolated with a 4th-order Butterworth filter. The signal within this band is then processed with a soft-clipper to gently introduce harmonic saturation reminiscent of vacuum tubes when driven.

---

**Algorithm 2** DSP Anchor Generation: Warmth

---

1: **Input:** Raw audio signal $x$
2: **Parameters:** Filter Order $N = 4$, Low Cutoff $f_{\text{low}} = 250\,\text{Hz}$, High Cutoff $f_{\text{high}} = 4\,\text{kHz}$
3: $x_{\text{mid}} \leftarrow \text{ButterworthBandpass}(x, N, f_{\text{low}}, f_{\text{high}})$ ▷ Isolate mid-band for warmth and presence.
4: $y_{\text{mid}} \leftarrow \tanh(1.1 x_{\text{mid}})$ ▷ Apply soft-clipping to emulate tube saturation.
5: $y \leftarrow y_{\text{mid}} + (\text{x} - x_{\text{mid}})$ ▷ Combine processed mid-band with original signal.
6: **Output:** Processed audio signal $y$

---

**Mastering (Layeredness)** To achieve a polished and layered master, we first apply loudness normalization, targeting -14 LUFS using the `pyloudnorm` library to meet modern streaming standards. Subsequently, a gentle spectral tilt is applied with a pivot frequency at 1kHz and a slope of -1.5 dB/octave. This common mastering technique subtly reduces high-frequency harshness and enhances low-frequency weight, improving the perceived depth and separation between instrumental layers.

---

**Algorithm 3** DSP Anchor Generation: Layeredness

---

1: **Input:** Raw audio signal $x$
2: **Parameters:** Target Loudness $L = -14\,\text{LUFS}$, Pivot Frequency $f_p = 1\,\text{kHz}$, Slope $S = -1.5\,\text{dB/octave}$
3: $x_{\text{norm}} \leftarrow \text{LoudnessNormalize}(x, L)$ ▷ Using the `pyloudnorm` library.
4: $y \leftarrow \text{SpectralTilt}(x_{\text{norm}}, f_p, S)$ ▷ Enhance low-end weight and reduce harshness.
5: **Output:** Processed audio signal $y$

---

### A.3 TRAINING AND INFERENCE PROCEDURES

To further elucidate our methodology and ensure the reproducibility of our work, this section provides detailed pseudocode for the two core processes of TimbrePalette: the model training loop and the style blending inference procedure. Each algorithm is preceded by a comprehensive description of its operational logic.

**Algorithm 1: TimbrePalette Training Procedure** The training of TimbrePalette is detailed in Algorithm 4. We employ an on-the-fly data generation strategy to create a diverse and virtually infinite training set. For each training step, we first load a full audio file from our training data paths. A short segment (e.g., 2 seconds) is then randomly cropped from this file to serve as the model input, $x_{segment}$. Concurrently, a target style $s$ is randomly selected from our set of defined Style Anchors. The corresponding pre-defined DSP function, $D_s$, is then applied to the input segment to generate the ground truth target audio, $y_s$. The chosen style index is converted into a one-hot vector, $v_s$, which acts as the conditional input for our model. The model $G_\theta$ performs a forward pass, taking both the audio segment and the style vector to produce the enhanced output $y'$. Finally, a loss is computed between the predicted audio and the ground truth, and the model's parameters $\theta$ are updated via backpropagation.

**Algorithm 2: Style Blending Inference Procedure** Algorithm 5 outlines the procedure for performing style blending during inference, a key feature of TimbrePalette. This process allows for the creation of novel hybrid timbres by navigating the continuous latent space learned by the model. The function requires a pre-trained model $G_\theta$, an input audio waveform $x$, the indices of two source styles ($s_A$ and $s_B$), and a blending factor $\alpha$ between 0 and 1. First, the style indices are converted into their respective one-hot vector representations, $v_{s_A}$ and $v_{s_B}$. The core of the technique is the linear interpolation of these two vectors to create a new, blended style vector, $v_{blend}$. This interpolated vector, which represents a point in the latent space between the two original styles, is then fed into the model alongside the input audio. The model processes these inputs and generates a single output waveform, $y_{blend}$, which perceptually embodies the mixed characteristics of the two source styles, determined by the factor $\alpha$.

---

**Algorithm 4** TimbrePalette Training Loop

---

**Require:** Model $G_\theta$, Optimizer $\theta_{optim}$, Loss Function $\mathcal{L}$, Training Data Paths $X_{train}$, DSP Style Anchors $\{D_s\}_{s=1}^3$
1: **for** each epoch **do**
2:     **for** each audio path $x_{path}$ in $X_{train}$ **do**
3:         $x_{full} \leftarrow \text{LoadAudio}(x_{path})$
4:         $x_{segment} \leftarrow \text{RandomCrop}(x_{full})$
5:         $s \leftarrow \text{RandomSelect}(\{1, 2, 3\})$
6:         $y_s \leftarrow D_s(x_{segment})$
7:         $v_s \leftarrow \text{OneHot}(s)$
8:         $y' \leftarrow G_\theta(x_{segment}, v_s)$
9:         $loss \leftarrow \mathcal{L}(y', y_s)$
10:        $loss.\text{backward}()$
11:        $\theta_{optim}.\text{step}()$
12:     **end for**
13: **end for**

---

---

**Algorithm 5** TimbrePalette Style Blending Inference

---

**Require:** Trained Model $G_\theta$, Input Audio $x$, Style A index $s_A$, Style B index $s_B$, Blend factor $\alpha \in [0, 1]$
1: $v_{s_A} \leftarrow \text{OneHot}(s_A)$
2: $v_{s_B} \leftarrow \text{OneHot}(s_B)$
3: $v_{blend} \leftarrow (1 - \alpha)v_{s_A} + \alpha v_{s_B}$
4: $y_{blend} \leftarrow G_\theta(x, v_{blend})$ **return** $y_{blend}$

---

## A.4 MODEL AND EXPERIMENTAL DETAILS

To ensure full reproducibility, this section provides exhaustive details on the final model architecture, training hyperparameters, datasets, and the computational environment used in our main experiments.

### A.4.1 FINAL MODEL ARCHITECTURE (CONDITIONALWAVEUNET-V2)

Our final model, `TimbrePalette` (`ConditionalWaveUNet-v2`), is a fully convolutional time-domain model. Its architecture, verified against the implementation in `sec8_train_conditional_model.py`, is detailed in Table 7. The model uses `ConvTranspose1d` for upsampling. The total number of trainable parameters is **1,454,241**.

### A.4.2 TRAINING HYPERPARAMETERS

All hyperparameters are sourced directly from our final training script, `sec8_train_conditional_model.py`.

- **Optimizer:** AdamW

- **Learning Rate:** $2 \times 10^{-4}$ (initial max learning rate)

- **Weight Decay:** $1 \times 10^{-5}$

- **Learning Rate Scheduler:** One-Cycle Learning Rate Scheduler (`OneCycleLR`)

- **Loss Function:** A combination of L1 loss in the time domain and L1 loss on the magnitude of the STFT (`StableLoss`).

- **STFT Parameters:** Hann window, FFT size of 2048, hop length of 512.

- **Batch Size:** 8

- **Epochs:** 50

- **Numerical Stability:** Automatic Mixed Precision (AMP) was enabled for all training runs.

Table 7: Architecture of our final `ConditionalWaveUNet` model. 'k' denotes kernel size, 's' denotes stride, and 'c' denotes channels. All convolutional layers use GELU activation.

| Layer Type | Layer Name | Output Shape Transformation |
|---|---|---|
| *Encoder* | | |
| Input | - | `[B, 1, 441000]` |
| Conv1d | `inc` (k=15, s=1) | `[B, 32, 441000]` |
| DownBlock | `d1` (k=15, s=2) | `[B, 64, 220500]` |
| DownBlock | `d2` (k=15, s=2) | `[B, 128, 110250]` |
| DownBlock | `d3` (k=15, s=2) | `[B, 256, 55125]` |
| *Style Conditioning* | | |
| One-hot Style Vector | - | `[B, 3]` |
| MLP | `style_mlp` | `[B, 256]` |
| Bottleneck | `bottleneck` | `[B, 256, 55125]` (element-wise add) |
| *Decoder* | | |
| UpBlock | `u1` (ConvTranspose1d) | `[B, 128, 110250]` (concat with `d2` output) |
| UpBlock | `u2` (ConvTranspose1d) | `[B, 64, 220500]` (concat with `d1` output) |
| UpBlock | `u3` (ConvTranspose1d) | `[B, 32, 441000]` (concat with `inc` output) |
| Conv1d | `outc` (k=1, s=1) | `[B, 1, 441000]` |
| Activation | `tanh` | `[B, 1, 441000]` |

### A.4.3 DATASET AND PREPROCESSING

- **Training Datasets:** We used the full development set from **DSD100** ("Dev" subset) and the full training set from **MUSDB18-HQ** ("train" subset). As confirmed by the log in `sec8`, this resulted in a total of 150 unique songs for training.

- **Test Datasets:** For the final objective evaluation in `sec9`, we used the official test sets from both **DSD100** ("Test" subset) and **MUSDB18-HQ** ("test" subset).

- **Audio Preprocessing:** All audio files were resampled to a target sample rate of 44100 Hz and converted to mono by averaging channels if necessary.

- **Data Segmentation:** During training, we used random 10-second segments cropped from the full audio files. For evaluation and inference, the first 10 seconds of each track were used.

### A.4.4 COMPUTATIONAL ENVIRONMENT

- **Hardware:** All experiments were conducted on a single CUDA-enabled NVIDIA GPU.

- **Software:** The framework was built using Python 3.10. Key libraries include PyTorch, torchaudio, librosa, and for the "mastering" style anchor, `pyloudnorm`.

- **Training Time:** Training the final model for 50 epochs on the full dataset took approximately **3.8 hours**.

## A.5 SUPPLEMENTARY VISUAL ANALYSIS

In this appendix, we provide the complete set of visual analysis figures for various audio samples from our test set. These figures complement the compact analytical figure presented in the main paper (Figure 2) by offering a more detailed comparison of our model's performance against both the `SpectrogramUNet` and `ConvTasNet` baselines across a diverse range of audio content.

For each audio sample analyzed, we present a pair of comprehensive analysis figures: one comparing our model against the `SpectrogramUNet` baseline, and the other against the `ConvTasNet` baseline. Each of these figures is a self-contained visual report, including: a 2x2 grid of spectrograms, a spectral difference plot, a frequency band energy comparison, and a numerical summary table. All spectrograms are focused on a representative 3-second segment to highlight fine details.

### A.5.1 ANALYSIS ON AI-GENERATED SAMPLE

Visualized comparison of TimbrePalette and baseline generated Guitar samples, As shown in Figure 4 and Figure 5.

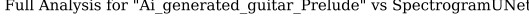
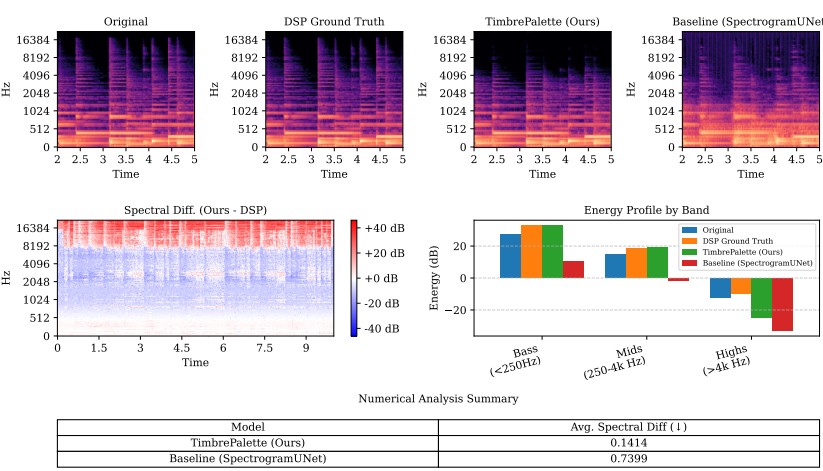

Figure 4: Full analysis for an AI-generated guitar sample (`Ai_generated_guitar_Prelude.wav`), comparing `TimbrePalette` with the `SpectrogramUNet` baseline. The figure contains a comprehensive visual report, including spectrograms, a spectral difference plot (Ours vs. DSP), a bar chart comparing energy by frequency band, and a numerical summary. The plots demonstrate our model's ability to correctly apply the "Fullness" style's spectral characteristics, closely matching the DSP ground truth, whereas the baseline fails to replicate the intended low-frequency boost.

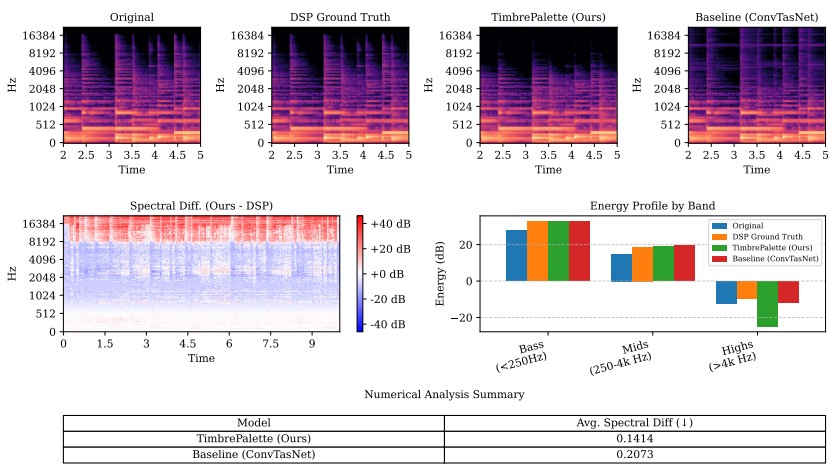

Figure 5: Full analysis for the same AI-generated guitar sample (`Ai_generated_guitar_Prelude.wav`), comparing `TimbrePalette` with the `ConvTasNet` baseline. The energy band plot and the low spectral difference value in the table highlight our model's superior accuracy in matching the DSP target's frequency profile compared to the strong time-domain baseline.

### A.5.2 ANALYSIS ON VOCAL SAMPLES

Visualized comparison of TimbrePalette and baseline generated Vocal samples, As shown in Figure 6 and Figure 7.

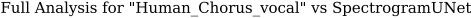
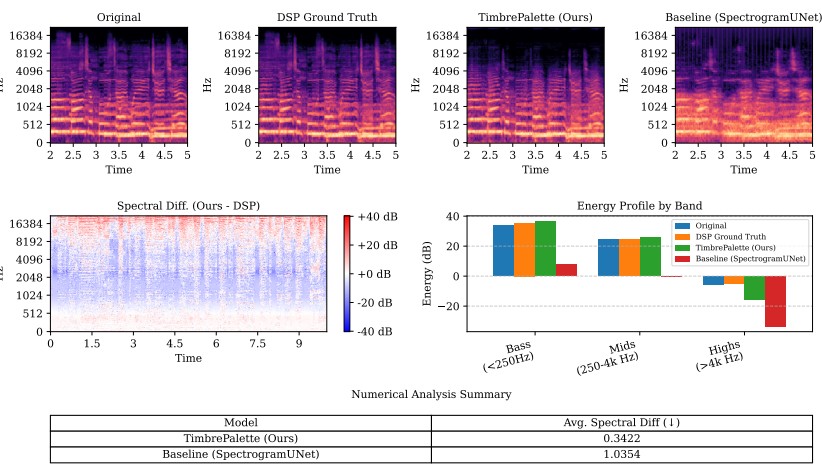

Figure 6: Full analysis for a chorus vocal sample (Human_Chorus_vocal.wav), comparing against the SpectrogramUNet baseline. The spectral difference plot (Ours vs. DSP) shows predominantly neutral colors, indicating minimal error for our model. In contrast, the baseline's large spectral difference and incorrect energy profile demonstrate its inability to handle this type of audio.

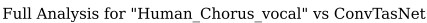
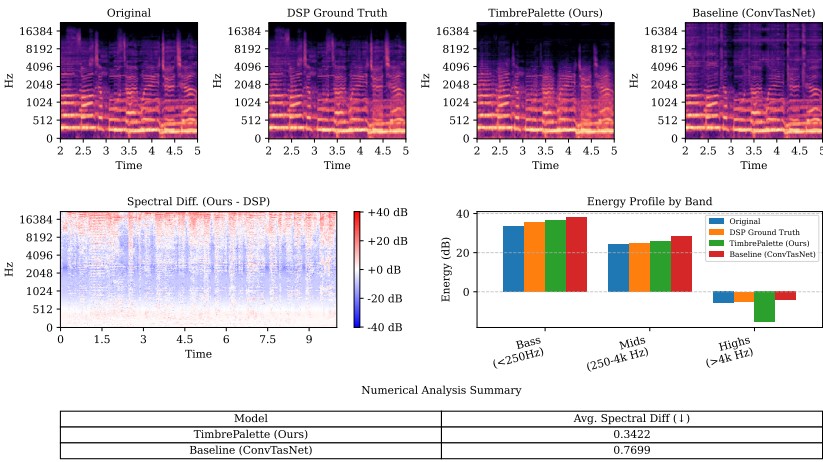

Figure 7: Full analysis for the chorus vocal sample (Human_Chorus_vocal.wav), comparing against the ConvTasNet baseline. This figure provides a direct comparison against a strong time-domain competitor, showcasing the superior performance of our proposed architecture on complex vocal material.

### A.5.3 ANALYSIS ON PIANO SAMPLES

Visualized comparison of TimbrePalette and baseline generated Piano samples, As shown in Figure 8 and Figure 9, Figure 10 and Figure 11, Figure 12 and Figure 13, .

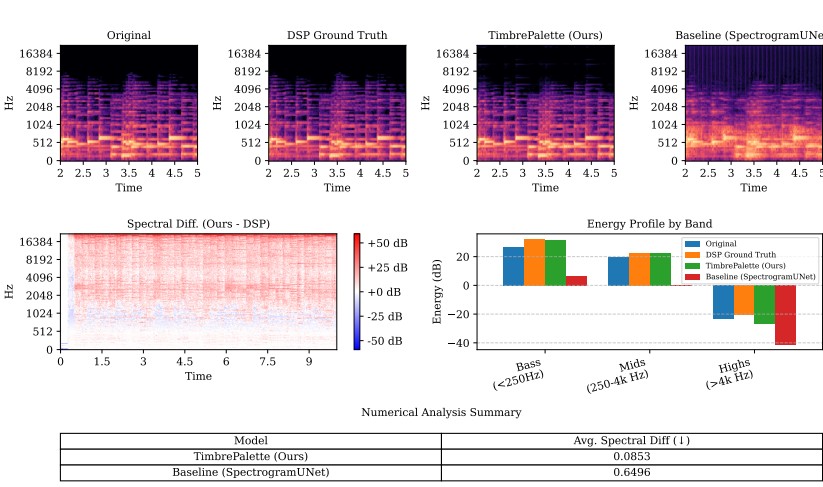

Figure 8: Full analysis for a piano solo sample (`human_pianoSolo1.wav`), comparing against the `SpectrogramUNet` baseline. The spectrograms and energy chart clearly show our model successfully adding fullness and body in the low-mid frequencies, closely mirroring the DSP ground truth.

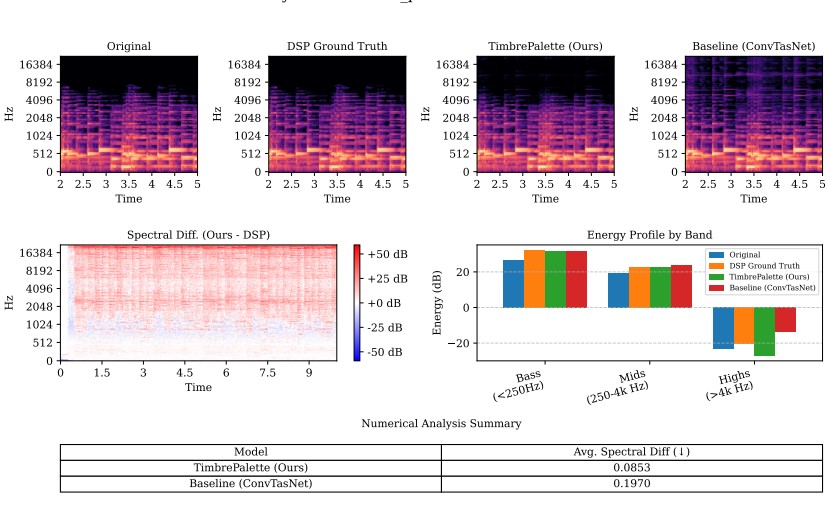

Figure 9: Full analysis for the first piano solo sample (`human_pianoSolo1.wav`), comparing against the `ConvTasNet` baseline. The numerical summary table quantifies our model's lower spectral difference, reinforcing the visual evidence of its higher fidelity.

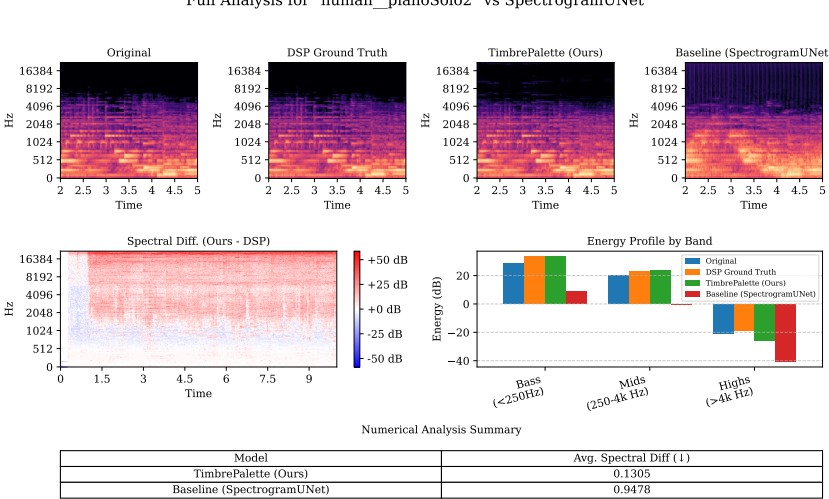

Figure 10: Full analysis on a second piano solo sample (`human_pianoSolo2.wav`) against the `SpectrogramUNet` baseline. This example further validates our model's consistent performance on acoustic instrument recordings.

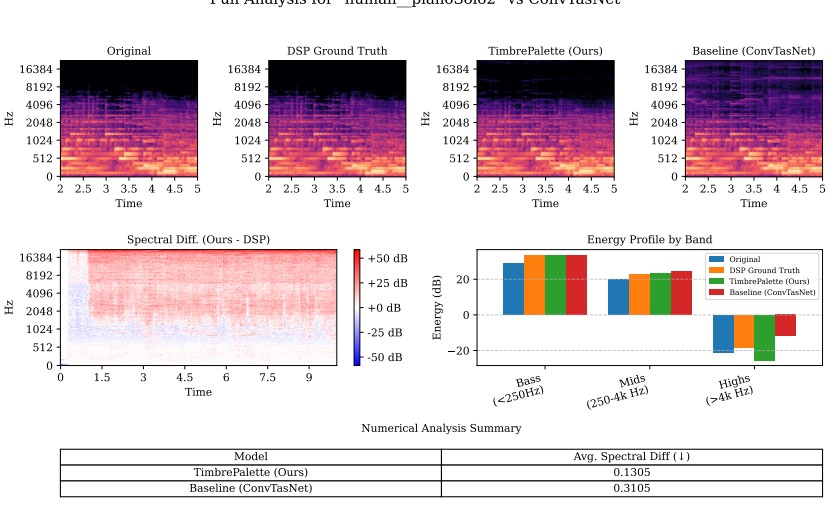

Figure 11: Full analysis on a second piano solo sample (`human_pianoSolo2.wav`) against the `ConvTasNet` baseline, confirming the performance gap between our specialized architecture and the SOTA source separation model on this task.

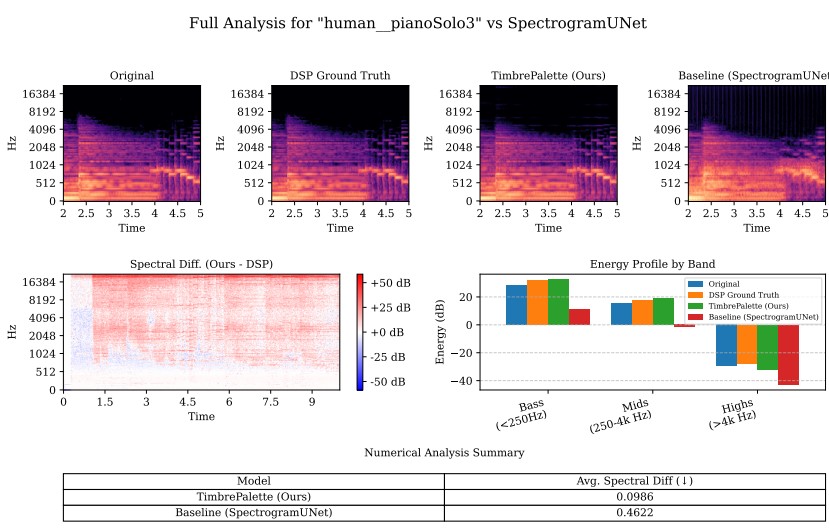

Figure 12: Full analysis on a third piano solo sample (`human_pianoSolo3.wav`) against the `SpectrogramUNet` baseline. The visuals consistently show the failure of the frequency-domain model and the success of our time-domain approach.

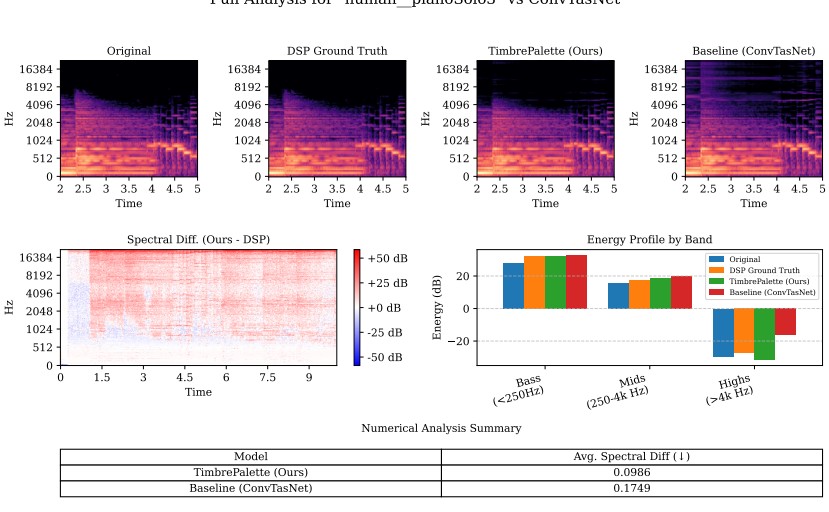

Figure 13: Full analysis on a third piano solo sample (`human_pianoSolo3.wav`) against the `ConvTasNet` baseline, providing another data point for the robustness of our results across different examples.

## A.6 USAGE OF LARGE LANGUAGE MODELS (LLMS)

In adherence to the conference guidelines, we wish to disclose the role that Large Language Models (LLMs) played in the preparation of this manuscript and during the research process. The LLM was utilized in two primary capacities: as a writing assistant for language refinement and as an interactive debugging tool.

- **Language Polishing of the Introduction:** The Abstract and Introduction section of this paper was polished on a sentence-by-sentence basis using an LLM. For each sentence, we provided the model with a prompt such as, "Please make the following sentence more concise, fluent, and easy to understand" to improve the overall clarity and readability of the text.

- **Interactive Debugging and Code Fixing:** Throughout the experimental phase of this research, we encountered numerous warnings, errors, and exceptions. To accelerate the debugging process, we provided the LLM with error messages, tracebacks, and the corresponding code snippets. We used prompts such as, "Please fix the errors in this code snippet based on the error message," to help identify the root causes of bugs and generate potential solutions.

The LLM served as a general-purpose assistance tool to enhance the quality of the writing and the efficiency of the coding workflow. The core research ideation, experimental design, and final analysis presented in this paper are the original contributions of the authors.

