# OpenReview forum: "TimbrePalette: A Controllable Multi-Style Generation Model for Timbre Enhancement"
_ICLR.cc/2026/Conference — ICLR 2026 Conference Withdrawn Submission_

### Official Review · Reviewer_ssc3 · 2025-10-30

**Soundness:** 1
**Presentation:** 2
**Contribution:** 1
**Rating:** 2
**Confidence:** 4

**Summary:**

The paper introduces **TimbrePalette**, an innovative, controllable multi-style timbre enhancement model built on a conditioned **Wave-U-Net** architecture. The model addresses the limitations of traditional Digital Signal Processing (DSP) effect chains, which lack content-awareness, and naive deep learning methods, which often suffer from training instability when imitating complex audio effects.

Features of the model include:

* **Training with Style Anchors:** The authors define three high-quality DSP algorithms representing distinct perceptual dimensions ("Fullness," "Warmth," and "Layeredness") to serve as "Style Anchors."
* **Timbre Model with Controllable Generation:** A single, unified TimbrePalette model is trained to learn the generation of enhanced audio corresponding to an explicit style command.
* **Latent Space Interpolation:** The model exhibits the ability to smoothly "blend" between the individual styles, indicating that it has learned a meaningful and continuous latent space of timbre aesthetics.

**Strengths:**

- explicit description of failed approaches
- Exploration of a new timbre manipulation approach

**Weaknesses:**

### timbre space too limited and missing motivation of the training strategy

If I have understood correctly, the timbre space is represented by means of three DSP algorithms that are moderately adaptive and somewhat nonlinear. The 3 algorithms are presented to the model in the form of a one-hot conditioning, where only one of the algorithms is active. During inference, the study evaluates a weighted combination of these individual effects. I do not understand how the model is supposed to learn how a combination of these effects should be performed.

### missing context and relevant references

The field of AI supported Audio Effects and automated mixing is quite active and broad. I don't quite understand why we need a rather detailed comparison of the C-U-Net, but I do not find more examples of the work in that area. I only give a few references that appear relevant to me (please note that I did not contribute to any of these papers!):

- V. Välimäki and J. D. Reiss. All about audio equalization: Solutions and frontiers. Applied Sciences, 6(5):129, 2016.
- Venkatesh, Satvik, David Moffat, and Eduardo Reck Miranda. "Word embeddings for automatic equalization in audio mixing." arXiv preprint arXiv:2202.08898 (2022). https://arxiv.org/pdf/2202.08898
- C. Peladeau and G. Peeters. Blind estimation of audio effects using an auto-encoder approach and differentiable digital signal processing. ICASSP, pages 856–860, 2024

### latent space interpolation experiment is not convincing

Concerning the latent space interpolation experiment, I would see two explanations of your observations. These small steps could mean that the model does hardly change anything (and so basically does not move), or as you say, that the model has learned a meaningful representation of the interpolated effects. The L1 distance measure is not informative, because it would change with a gain or STFT normalization

### motivation of the subjective test is unclear.

If I understand the perceptual test correctly, then the authors selected weighting coefficients that were used to generate sound effects that were then evaluated by listeners, who were asked to select the preferred effect. Now this demonstrates that listeners preferred one or another of these weightings, but does not inform us about the successful interpolation in the effect timbre space. A question that might have been tested would have been whether listeners prefer the original DSP effects, or prefer the reproductions with the model, or even prefer the weighted effects.

### subjective test results do not show a significant difference.

Looking into your table 6, I do not see a significant preference. For the first 3 lines, MOS score difference is in the order of 0.1, whole STDDEV is in the order of 0.9. In fact, I listened to a few of your examples and could, myself, not hear a difference between all the generated sounds. I could hear a difference compared to the original sound, though. While I may not have the best ears, I asked others to tell me where they can hear a difference, and the answer was mostly negative. This short expêreince is fully in line with the results you report. Even if there were a preference, this would not mean much for the present study (see above).

**Questions:**

### A few thoughts:

As you say, the problem you pose is ambitious. Creating a model that allows generating a weighted combination of multiple effects is not easy. I would feel that the problem is ill-defined. In the real world, there are numerous ways you may create mixtures of these effects: You can apply one after the other and control the effect degree (stronger saturation, stronger equalization). In that case, the result will depend on the order of the effects applied. Given that you do not select the order, the model will choose freely. Now your model may not apply these effects in sequence, but in parallel, and then create a linear mixture of the three independent results. If you want this situation, you don't really need to train a model. You have your individual, well-thought-through DSP effects; apply them individually and produce a weighted mix. In that case, you have the weighting factors, and you could learn to reproduce them with the model, but in that case there are no further non-linearities involved, besides the ones you have put into the DSP effect, so you don't really need any further model and training.

Given that you do not guide or constrain the solution space, the model will create a random combination, probably not one that is really achievable with a combination of your three DSP effects. The best way to evaluate all this remains to be seen, but my opinion is that the evaluation that is proposed here, does not do it.

---

### Official Review · Reviewer_26KX · 2025-10-31

**Soundness:** 2
**Presentation:** 3
**Contribution:** 2
**Rating:** 2
**Confidence:** 2

**Summary:**

The paper introduces TimbrePalette, using “ConditionalWaveUNet-v2” for controllable timbre enhancement. They define three style anchors(Fullness, Warmth, Layeredness) and trains a single model to map raw audio to these targets, with a simple bottleneck additive conditioning. Experiments use 2-s clips sampled from MUSDB18-HQ and DSD100 with on-the-fly DSP targets; metrics are STFT distance and SI-SNR; subjective tests involve 12 listeners. Reported results show gains over spectrogram U-Net, a non-conditioned variant, and Conv-TasNet; ablations suggest upsampling choice and conditioning matter.

**Strengths:**

1. Reproducible DSP anchors are spelled out (algorithms for Fullness, Warmth, Layeredness).
2. Consistent objective improvements over baselines; ablations diagnose failure without conditioning and with ConvTranspose.
3. Results outperform baseline models.

**Weaknesses:**

1. Novelty seems to be only about the style anchors and the ability to interpolate between styles.
2. Targets are self-defined DSP chains. Because “ground truth” is the authors’ DSP, success can amount to learning those fixed chains. That is closer to emulation than enhancement quality in the wild. There is no comparison to professional mastering chains, commercial plugins, or human producers.
3. Metrics are weak proxies. STFT-L1 and SI-SNR measure closeness to the DSP target, not human-perceived timbral quality, mix translation, or artifact perception; subjective evaluation is small (12 listeners) and limited to a few blends, with modest scores and no significance tests.
4. Claims of a “continuous aesthetic space” are under-evidenced. The “style blending” analysis is based on distance monotonicity and small-N listening; no latent traversal visualizations, disentanglement tests, or robustness checks across content types are provided.

**Questions:**

1. Why not compare against modern latent-codec (e.g., SoundStream-class) or diffusion enhancement models with conditioning? Even a lightweight diffusion baseline would strengthen claims.
2. Evaluation for only 2 seconds of audio seems to be too short. Is it possible to expand?
3. There might be some problems in the demo website. I can not download parts of the samples.
4. Is style blending the only advantage of training the generative model? Or why not just apply audio with the DSP algorithms?

---

### Official Review · Reviewer_e6ae · 2025-10-31

**Soundness:** 1
**Presentation:** 1
**Contribution:** 1
**Rating:** 0
**Confidence:** 5

**Summary:**

The paper presents a black-box approach to model 3 different types of “timbre” modeled with DSP.

**Strengths:**

couldn’t think of anything

**Weaknesses:**

I’d like to reject all the authors’ claims:

- **A Systematic Problem Analysis and Solution**
    - The reviewer strongly disagree with the claim: “we are the first to systematically document and solve the stability challenges prevalent in complex end-to-end audio enhancement tasks.” Refer to the missing reference part below.
- **An Innovative Conditioning Paradigm**
    - There are already many works on style transfer of audio effects that can perform more than 3 different classes. Refer to the missing reference part too.
- **A High-Performance Core Model**
    - Can we say Conv-TasNet is the SOTA for this task? Unless the authors can do an extensive research like Comunità et al, this is very hard to be justified. (M. Comunità, et al., "Differentiable black-box and gray-box modeling of nonlinear audio effects." Frontiers in Signal Processing, 2025)
- **A Successful Exploration of the Latent Space**
    - I’m not convinced with this statement at all.


**missing reference**

- number of references lacks a lot in general
- line 39-41: why? reference?
- line 42: why and only Demucs specifically?
    - line 42-43: is the degradation caused by separation model or the generative model?
- second paragraph of Introduction: strongly recommend doing proper literature search

    [1] S. Grishakov, et al., “Matchering: Audio matching and mastering python library,” https://github.com/sergree/matchering.

    [2] J. Sterne and E. Razlogova, “Machine learning in context, or learning from landr: Artificial intelligence and the platformization of music mastering,” Social Media+Society, vol. 5, no. 2, p. 2056305119847525, 2019.

    [3] M. A. M. Ramírez, et al., “Differentiable signal processing with blackbox audio effects,” in IEEE ICASSP 2021.

    [4] J. Koo, et al. “End-to-end music remastering system using self-supervised and adversarial training,” in IEEE. ICASSP 2022.

    [5] J. Koo, et al. “ITO-Master: Inference-time optimization for audio effects modeling of music mastering processors,” In ISMIR 2025.

    [6] C. Steinmetz, et al., “ST-ITO: Controlling audio effects for style transfer with inference-time optimization,” in ISMIR 2024.

    [7] C.-Y. Yu, et al., “Improving inferencetime optimisation for vocal effects style transfer with a gaussian prior,” in WASPAA 2025.

    [8] Hayes, Ben, et al., "Audio synthesizer inversion in symmetric parameter spaces with approximately equivariant flow matching." in ISMIR 2025.

    and many more…

- As I read the manuscript, I feel most sentence’s claim lacks reference. I recommend the authors to use proper references to claim each point or to refer previous method correctly.


Other comments:

- I couldn’t listen to (download) the “Enhanced” examples of Part 1 at the time of review.
- The model is trained with already processed files. What makes it different to apply audio effects to these already processed files?
- use \citep{} instead of \cite{} for referencing with brackets to enhance readability

**Questions:**

refer to weakness

---

### Official Review · Reviewer_9UGu · 2025-11-01

**Soundness:** 2
**Presentation:** 3
**Contribution:** 2
**Rating:** 4
**Confidence:** 4

**Summary:**

This paper proposes a conditioned Wave-U-Net architecture as an audio timbre enhancement model. The objective is to define a stable generative model that has the capabilities of DSP audio-enhancement plugins, while providing more flexibility and adaptation to the audio content. The main contribution is a robust training framework and a conditional time-domain Wave-U-Net (TimbrePalette) trained to reproduce three DSP-defined “Style Anchors” (Fullness, Warmth, Layeredness). The model conditions on one-hot style vectors projected by an MLP, and demonstrates continuous style blending via linear interpolation in style space.

**Strengths:**

The approach is clearly presented and reproducible. The task is relatively original within neural audio effects and clearly has practical value. The method successfully combines a generative audio architecture with style conditioning and DSP-based ground truths. The authors show objective improvements over strong baselines and present a style blending analysis, which demonstrates that the latent space is continuous, enabling creative practical blending of timbre styles using a single model instead of 3 DSP plugins. A small subjective listening test also supports the perceptual plausibility of the style blending.

**Weaknesses:**

Given the paper’s stated goals of efficiency, quality and user-friendliness, the experimental validation of these three aspects is incomplete:
- The efficiency of the approach (claimed in the abstract on line 30 and conclusion on line 485) is not explicitly shown (compared to applying DSP plugins). Please report inference speed and memory usage required to generate samples on CPU and GPU.
- For the audio quality, although the comparison against other models is sufficient to validate the approach, a complementary subjective listening test to assess the quality of the generated audio compared to DSP ground truths would have better validated the perceptual quality relative to them.
- Finally, for the user-friendliness, while I agree that combining 3 models into one enables a more practical and creative exploration of the latent space, it is not completely clear if using a generative model provides anything else in this context (compared to a chain of DSP plugins). The DSP plugins may have more parameters to calibrate but it seems that they were fixed to specific values during training.
This leads to my main criticism of the approach. The model is trained to approximate explicit DSP operations with the same parameters for all audio inputs. Even if the possibility of blending those DSP plugins at inference within a continuous latent space has practical creative value, it is not clear to me that the generative modeling approach represents a considerable advantage over combining those ground truth DSP operations. To be clearer, could you provide perceptual evidence that the outputs possess novel timbres, or are they simply a smooth approximation of interpolated DSP parameter settings?
The subjective listening test could have also compared against the DSP ground truths to assess if listeners prefer the style blending compared to pure single-style ground truths.

**Questions:**

- In the intro, you present DSP plugin chains as 'unable to intelligently adapt to varying audio content' (lines 44-45). How does your approach differ from that if you are using these DSP plugins as ground truths? Apart from the practical advantage of relying on a single model instead of calibrating 3 independent plugins, is there really an advantage in the generative possibilities of the presented approach?
- If these 3 DSP plugins can be combined (without a generative model) as a simple processing chain, could you compare those outputs to your style blended generated samples (subjective test)? On top of the practical (creative) aspect, this could also prove the high-quality claim of the paper.

---

### Note · Authors · 2025-11-12

**Comment:**

Dear Program Chair, Area Chairs, and Reviewers,

We are deeply grateful for your thorough review and invaluable feedback on our submission. We sincerely appreciate the time, expertise, and constructive criticism you dedicated to evaluating our work. Your insightful comments have significantly clarified areas requiring refinement, and we fully concur with your suggestions for strengthening the technical rigor and clarity of the paper.

In light of your recommendations, we are committed to revising the manuscript comprehensively to address all points raised. The revised version, incorporating every suggested improvement, will be submitted within the standard revision period.

Thank you for your exceptional guidance, which has been instrumental in elevating the quality and impact of our research. We are truly honored to have your expertise inform this important work.

With utmost respect

**Withdrawal Confirmation:**

I have read and agree with the venue's withdrawal policy on behalf of myself and my co-authors.